# Computational Methods for Understanding the Selectivity and Signal Transduction Mechanism of Aminomethyl Tetrahydronaphthalene to Opioid Receptors

**DOI:** 10.3390/molecules27072173

**Published:** 2022-03-28

**Authors:** Peng Xie, Junjie Zhang, Baiyu Chen, Xinwei Li, Wenbo Zhang, Mengdan Zhu, Wei Li, Jianqi Li, Wei Fu

**Affiliations:** 1School of Pharmacy, Fudan University, Shanghai 201301, China; 15111030005@fudan.edu.cn (P.X.); 21211030014@fudan.edu.cn (J.Z.); 20211030016@fudan.edu.cn (B.C.); 19211030070@fudan.edu.cn (X.L.); zhumengdan@gmail.com (M.Z.); wei-li@fudan.edu.cn (W.L.); 2Novel Technology Center of Chemical Pharmaceutics, Shanghai Institute of Pharmaceutical Industry Co., Ltd., China State Institute of Pharmaceutical Industry Co., Ltd., Shanghai 201203, China; zhangwenbo_2021@163.com

**Keywords:** μ-opioid receptor, δ-opioid receptor, κ-opioid receptor, activation mechanism, selectivity, molecular dynamics simulation, signal transduction mechanism

## Abstract

Opioid receptors are members of the group of G protein-couple receptors, which have been proven to be effective targets for treating severe pain. The interactions between the opioid receptors and corresponding ligands and the receptor’s activation by different agonists have been among the most important fields in opioid research. In this study, with compound **M1**, an active metabolite of tramadol, as the clue compound, several aminomethyl tetrahydronaphthalenes were designed, synthesized and assayed upon opioid receptors. With the resultant compounds **FW-AII-OH-1** (*K*i = 141.2 nM for the κ opioid receptor), **FW-AII-OH-2** (*K*i = 4.64 nM for the δ opioid receptor), **FW-DI-OH-2** (*K*i = 8.65 nM for the δ opioid receptor) and **FW-DIII-OH-2** (*K*i = 228.45 nM for the δ opioid receptor) as probe molecules, the structural determinants responsible for the subtype selectivity and activation mechanisms were further investigated by molecular modeling and molecular dynamics simulations. It was shown that Y^7.43^ was a key residue in determining the selectivity of the three opioid receptors, and W^6.58^ was essential for the selectivity of the δ opioid receptor. A detailed stepwise discovered agonist-induced signal transduction mechanism of three opioid receptors by aminomethyl tetrahydronaphthalene compounds was proposed: the 3–7 lock between TM3 and TM7, the DRG lock between TM3 and TM6 and rearrangement of I^3.40^, P^5.50^ and F^6.44^, which resulted in the cooperative movement in 7 TMs. Then, the structural relaxation left room for the binding of the G protein at the intracellular site, and finally the opioid receptors were activated.

## 1. Introduction

Opioid receptors (ORs) are important members of the type A (rhodopsin-like) G protein-coupled receptors (GPCRs) and have been considered to be potential drug targets for the treatment of moderate-to-severe pain, depression, affective disorders, irritable bowel syndrome and drug abuse [1,2,3,4].

Three main subtypes of ORs exist in the human body, namely the μ (MOR), δ (DOR) and κ (KOR) opioid receptors [5]. The majority of clinically used opioid analgesics, including morphine, pethidine, tramadol and oxycodone (Figure 1), selectively target the MOR. Their usage is restricted by side effects including addiction, tolerance, constipation, itching and respiratory depression [6]. These well-established adverse reactions are believed to be a consequence of MOR stimulation, and a biased signaling component was proposed to be involved in this process, which led to the development of opioids with potential functional selectivity [7,8]. The signal-biased opioids like **TRV130** (Oliceridine) and **PZM21** were shown to have better behaviors on adverse effects [9,10,11,12], and **TRV130** was approved by the Food and Drug Administration (FDA) for the management of severe acute pain in adult patients on 7 August 2020. However, **TRV130** also has the risks of addiction, abuse and misuse of opioids. Furthermore, despite a distinct safety profile in contrast to typical opioids, opioid-like substances targeting KOR and DOR receptors were also proposed to be potentially useful as clinical analgesics and antidepressants and for the treatment of irritable bowel syndrome (IBS). **ADL-5859** [13] and **ADL-5747** [14] are two selective DOR agonists which were tested in clinical phase II for the treatment of pain. Nalfurafine was the first to launch a selective KOR agonist for the treatment of pruritus in 2009 [15]. Eluxadoline was a μ/κ opioid dual agonist marketed for the treatment of IBS in 2015 [16]. Additionally, opioid antagonists such as naloxone and naltrexone were used to treat opioid abuse, which did not have the side effects related to the MOR agonism. Some scientists make many efforts to develop newly selective MOR antagonists or μ/κ opioid receptor dual ligands for the treatment of opioid use disorder (Figure 2) [17,18,19], and NFP could produce significantly fewer withdrawal symptoms than naloxone at similar doses in vivo.

Tramadol is an important opioid that has been clinically available over 30 years [20]. However, a few studies were conducted on tramadol derivatives due to its moderate analgesic activity. Oswald et al. [21] incorporated a benzene ring on positions 5 and 6 of tramadol and obtained aminomethyl dihydronaphthalene derivatives (Figure 2), which are selective DOR ligands, and the aminomethyl tetrahydronaphthalene derivatives were developed by Fu [22]. Thus, a potent DOR agonist **FW-AII-OH-2** and a moderately active KOR agonist **FW-AII-OH-1** was discovered (Figure 3). Due to the dominant pharmacological profiles, aminomethyl tetrahydronaphthalene was selected as the scaffold to provide new evidence to the molecular mechanism of tramadol and its structurally related analogs binding to different opioid receptors. They can be used as probes to investigate the structural determinants responsible for the subtype selectivity of these compounds and the activation mechanisms for opioid receptors. It is helpful to design and identify novel potent selective opioid ligands from aminomethyl tetrahydronaphthalenes. However, there is no computational study on the molecular mechanism of the binding mode, selectivity or activation of these analogs. In order to thoroughly clarify this issue, more aminomethyl tetrahydronaphthalene probes were designed, synthesized and tested for their activity against opioid receptors in this work. Finally, a series of computational methods were integrated to elucidate the molecular mechanism of activity, selectivity and signal transduction of these probes.

## 2. Results and Discussion

### 2.1. Chemical Design

The phenolic hydroxyl and *N*,*N*-dimethyl components were considered the essential group (“message” part) for tramadol and the structure-related analogs to bind with the opioid receptors, while the biphenyl group served as the “address” part (Figure 3a). In our previous work [22], different substituents were introduced to the biphenyl group (“address” part) to reveal its function. With **M1** as the starting compound, a benzene ring was added, and **FW-1** was synthesized, which was found to not be active for the three receptors. Alternatively, a biphenyl group was introduced to afford **FW-AII-OH-1** and **FW-AII-OH-2** (Figure 3b), which were identified as selective agonists of KOR and DOR, respectively. Since a hydrophobic pocket formed by Trp^6.48^ and Tyr^7.43^ (Ballesteros–Weinstein numbering [23]) existed around the “message” part [24], and the phenylethyl was introduced to the protonated nitrogen atom to obtain **FW-DⅢ-OH-2**, which increased the size of the “message” part. Whether such structural changes shared a similar pharmacological function with that of **FW-AII-OH-1** and **FW-AII-OH-2**, **FW-DⅢ-OH-2** was found to have moderate affinity with DOR. Therefore, the structural motif of **FW-All-OH-2** and **FW-DⅢ-OH-2** was combined to yield **FW-DI-OH-2**, which demonstrated a high affinitive for DOR in vitro. These molecules had little structural differences on the “message” or “address” parts but activated different subtypes of opioid receptors. Therefore, they could be used as probes to disclose the selectivity and activation mechanisms of ligands on the three subtypes of opioid receptors.

### 2.2. Chemical Synthesis

The synthesis of **M1**, **FW-AII-OH-1** and **FW-AII-OH-2** was reported in our previous paper [22], and the synthetic routes for the preparation of **FW-DI-OH-2** and **FW-DIII-OH-2** are shown in Figure 1, indicating that the synthesis of the two compounds started from 6-hydroxy-1-tetralone (compound **1**). Starting material **1** was treated with 2-bromo-2-methylpropanamide under basic conditions before hydrolysis to yield **2**. Then, this intermediate was further diazotized with copper bromide to afford **3**. Compound **3** was treated by a Suzuki reaction with phenylboronic acid and (2-methoxyphenyl) boronic acid to obtain compounds **4** and **7**, which were subsequently treated by a Mannich reaction to afford compounds **5** and **8**. Thereafter, these intermediates were transformed to **FW-DIII-OH-2** and **FW-DI-OH-2** by Grignard reactions, respectively.

### 2.3. In Vitro Activities

The activities of the probe compounds in vitro are summarized and listed in Table 1 for the binding affinities and Table 2 for their abilities to stimulate [^35^S]GTPγS binding.

The introduction of substituents (-F, -Cl, -CH_3_, -CF_3_ and -OCF_3_) at the ortho-position on the biphenyl group dramatically enhanced both the affinity and selectivity toward DOR, while the substitutions at the meta- or para-positions did not exhibit activity toward the three opioid receptor subtypes. Replacing one of dimethyl groups on the nitrogen atom with phenylethyl acquired the best activity for DOR, and other substituent groups such as benzyl, phenylpropyl and phenylbutyl would all make the activity of DOR decrease or disappear.

### 2.4. Results of Molecular Dynamics Simulations

The configuration of the two chiral centers is critical to confer the opioid-like activities for tramadol and the structurally related analogs. Theoretically, there should be four diastereomers for tramadol. Nevertheless, only trans-configuration enantiomers (e.g., 1*R*,2*R* or 1*S*,2*S*) were isolated during the synthesis routine for tramadol and its analogs, which were considered a consequence of an unfavorable steric repulsion between these two chiral centers during the Grignard reactions. Moreover, two pairs of racemic isomers should have been isolated by flash chromatography if they were present. In fact, only a racemic mixture composed of (1*R*,2*R*) isomer and (1*S*,2*S*) isomer was produced, purified and characterized for each compound in this study.

Additionally, the pharmacological characteristics were profiled well for the tramadol isomers. The (1*R*,2*R*) isomer of tramadol and its metabolite **M1** were identified as selective MOR agonists, while the (1*S*,2*S*) isomer was an inhibitor for noradrenergic reuptake, which is irrelevant to the opioid receptor system [25]. Therefore, it is reasonable to assume that the opioid-like activities were produced by the (1*R*,2*R*) isomer, while the other isomer was inactive as an opioid ligand. In addition, in our previous publication [22] related to this study, the two isomers of compound **6j** (corresponding to the compound **FW-AII-OH-2** in this manuscript) were separated from each other by chiral HPLC columns to afford compound (+)-**6j** and (−)-**6j**. Furthermore, only compound (−)-**6j** demonstrated highly selective and potent DOR activities in vitro as confirmed by binding and functional assays, whereas compound (+)-**6j** was inactive in all biological assays. This provides additional evidence for our hypothesis.

Thus, the biological activities described for the racemic isomers could be considered a rough representation for those (1*R*,2*R*) isomers. It might be appropriate to choose the (1*R*,2*R*) isomers of **FW-AII-OH-1**, **FW-AII-OH-2**, **FW-DI-OH-2** and **FW-DIII-OH-2** for further molecular modeling (Figure 4).

### 2.5. Selectivity of Probe Molecules

In our simulations, cluster analyses were performed in every system, and the representative conformations were used to explore the mechanism of the ligands’ selectivity for MOR, DOR and KOR. The binding modes are shown in Figure 5. All the compounds could form a salt bridge with D^3.32^ through the protonated nitrogen atom, and the phenolic hydroxyl group interacted with H^6.52^ via two water molecules, forming a hydrogen bond network in the active sites of the OPs. However, few differences between their structures led to different selectivity for the three receptors. In the **M1**-MOR system, the benzene ring of M1 formed a π-π stacking interaction with Y^7.43^ (Table 3) (Figure 5a), the benzene ring in tetrahydronaphthalene of **FW-AII-OH-1** formed a π-π stacking interaction with Y^7.43^ for KOR, and I^6.55^ and I^7.39^ formed hydrophobic interactions with the diphenyl component of **FW-AII-OH-1** (Figure 5c). Similarly, the benzene ring in phenylethylamine of **FW-DI-OH-2** and **FW-DIII-OH-2** also formed a π-π stacking interaction with Y^7.43^ for DOR (Figure 5d,e). Meanwhile, in terms of the **FW-AII-OH-2**-DOR complex, the side chain of Y^7.43^ flipped and formed a hydrogen bond with the nitrogen atom of **FW-AII-OH-2** (Figure 5b). In terms of the **FW-AII-OH-2**-DOR, **FW-DI-OH-2**-DOR and **FW-DIII-OH-2**-DOR systems, V^6.55^, F^6.54^, I^7.39^ and W^6.48^ formed a hydrophobic interaction with the ligands, and the benzene of the diphenyl component could form a π-π stacking interaction with W^6.58^ (Figure 5b,d,e), which was essential for DOR binding [26]. With regard to **FW-DI-OH-2** and **FW-DIII-OH-2**, except for the same interactions between Y^7.43^ and W^6.58^, in the **FW-DI-OH-2**-DOR system, the nitrogen atom in the ligand formed the salt bridge with D^3.32^, and meanwhile, the phenolic hydroxyl group formed a hydrogen bond with D^3.32^ (Figure 5e), while there was only a salt bridge interaction between **FW-DIII-OH-2** and D^3.32^ (Figure 5d). These can explain why the binding affinity of **FW-DIII-OH-2** was weaker than that of **FW-DI-OH-2**.

It was proposed that the substituent group in the “address” part would enhance the selectivity and affinity for DOR and KOR. There was another bulky group in **FW-AII-OH-1** and **FW-AII-OH-2** compared with the “address” part of **M1**. Because of the introduction of large bulky substituents, the binding poses of the ligand endured some significant changes. Compared with the binding pose of **M1** in MOR, the nitrogen atom in **FW-AII-OH-2**, and the tetrahydronaphthalene in **FW-AII-OH-1**, steric clashes occurred with D^3.32^ (Figure 6a,b). According to the alignment of the key amino residues around the binding sites of MOR, DOR and KOR (Figure 7), the significant difference was in position 6.58, lysine (K) in MOR, tryptophan (W) in DOR and glutamic acid (E) in the KOR. Lysine and glutamic acid were hydrophilic and polar amino acids, but tryptophan was a hydrophobic and nonpolar amino acid. The 2-Methoxy diphenyl component in **FW-AII-OH-2** is a hydrophobic substituent, and the lysine in MOR and glutamic acid in DOR should make this “address” part less favorable for forming hydrophobic interactions with such a hydrophobic substituent in **FW-AII-OH-2** (Figure 6c).

### 2.6. The Salt Bridges between D^3.32^ and the Amine Group

Many classical opioid receptor ligands could form a salt bridge between the residue D^3.32^ and the protonated amine group of ligands [27,28,29,30]. In our work, salt bridge interactions in the **M1**-MOR, **FW-AII-OH-2**-DOR and **FW-AII-OH-1**-KOR systems were found, as indicated in the initial docking poses. During the MD simulations, the salt bridges were exited during the whole MD simulation processes (see Appendix A). This result illustrates that in our MD systems, the ionic interaction between D^3.32^ and the ligand was the initial and most important driving force for the ligand’s binding.

### 2.7. The 3–7 Lock

The 3–7 lock, which is the hydrogen bond formed between D^3.32^ in TM3 and Y^7.43^ in TM7, was suggested to play an important role in the signal transduction of the opioid receptors [31]. During our simulations, the distances of the hydrogen bonds between D^3.32^ and Y^7.43^ in each system were monitored and compared with those of opioid receptors in an inactive state. In terms of inactive MOR (4DKL) and DOR (4N6H) systems, the 3–7 lock existed (Figure 8b,d), and the distances of the two systems were both 2.6 Å. In the case of an agonist-bound MOR complex, the 3–7 lock was broken, and the distance stayed ca. 6.9 Å (Figure 8a). Meanwhile, in the case of **FW-AII-OH-2**-DOR, the 3–7 lock was maintained in the simulation’s time, and the distance stayed ca. 2.7 Å (Figure 8c) and did not happen to break like that of **M1**-Md, a different form. The 3–7 lock in inactive KOR (4DJH) did not exist (Figure 8f). In the case of the **FW-AII-OH-1**-KOR complex, the 3–7 lock was formed, and the distance stayed at about 2.6 Å (Figure 8e), which was the same as that in inactive MOR (4DKL) and DOR (4N6H). The monitored distances between D^3.32^ and Y^7.43^ in all complexes are shown in Figure 9.

The 3–7 lock showed different behavior in the activation process of MOR, DOR and KOR induced by **M1**, **FW-AII-OH-2** and **FW-AII-OH-1**. In terms of the MOR-M1 system, D^3.32^ and Y^7.43^ maintained hydrogen bonding in the inactive state. When M1 bound to MOR, it was inserted into the small space between D^3.32^ and Y^7.43^. The protonated nitrogen atom of **M1** formed a hydrogen bond with D^3.32^, which prevented the interaction between D^3.32^ and Y^7.43^, and Y^7.43^ was far away from D^3.32^ (Figure 8a,b). Then, the 3–7 lock was broken from an inactive state to an active one. In terms of the DOR-**FW-AII-OH-2** system, when **FW-AII-OH-2** combined with DOR, **FW-AII-OH-2** formed bi-dentated hydrogen bonds with D^3.32^ and Y^7.43^, which formed a stable triplet that stabilized the hydrogen bond between D^3.32^ and Y^7.43^ (Figure 8c,d). Therefore, the 3–7 lock went from existing in an inactive state to an active state. In terms of KOR- **FW-AII-OH-1**, when **FW-AII-OH-1** bound to KOR, the protonated nitrogen atom of **FW-AII-OH-1** was just located below D^3.32^ and Y^7.43^, and it formed an interaction with D^3.32^ and pulled D^3.32^ toward Y^7.43^, promoting the formation of a hydrogen bond interaction between D^3.32^ and Y^7.43^ (Figure 8e,f) such that the 3–7 lock was formed in the active state.

### 2.8. The DRY Lock (Connection between TM3 and TM6)

The conserved E/DRY motif located at the intracellular end of TM3 played an important role in maintaining GPCRs in the inactive state [30]. Taking rhodopsin as an example, an ionic interaction between R^3.50^ and E^6.30^ in TM6 stabilized TM6 in the inactive state [28]. There was no acidic amino acid residue at position 6.30 in the ORs to form a similar salt bridge with R^3.50^ to make TM6 stable. Instead, T^6.34^ interacted with R^3.50^ to maintain the protein in the inactive conformation (Figure 10b,d,f). In our simulations, the R^3.50^–T^6.34^ hydrogen bond was never found to be formed (Figure 10a,c,e), which might be attributed to the break of the intra-helical salt bridge between R^3.50^ and D^3.49^ that enables R^3.50^ to interact with Y^5.58^. This disruption was maintained in the whole simulation process (Figure 11). The interaction between R^3.50^ and Y^5.58^ made Y^5.58^ form a water bridge-mediated hydrogen bond network with Y^7.53^, which was a part of the conserved NPxxY motif in the intracellular end of TM7 [32].

### 2.9. The Rearrangements of I3.40, P5.50 and F6.44

The outward motion of the intracellular end of TM6 is a major conformational change exhibited by most GPCRs, including opioid receptors [33,34], which appears to be linked to the conformational change of residues in the orthosteric binding site. For opioid receptors, the rearrangements of the triad of conserved residues I^3.40^, P^5.50^ and F^6.44^ happened just below the orthosteric binding site [30]. Therefore, we compared the conformation of these three amino acids between the ligand-free protein and the agonist-binding one (Figure 12).

The rearrangement of I^3.40^, P^5.50^ and F^6.44^ was associated with a counter-clockwise rotation (extracellular viewer) and outward movement of the cytoplasmic end of TM6. In this way, the G protein could bind with the intercellular ends of the opioid receptors and activate their downstream pathways.

### 2.10. The Movements of Transmembrane Helixes

Followed by the rearrangements of the conserved residues I^3.40^, P^5.50^ and F^6.44^ that laid just below the orthosteric binding site, the transmembrane helix would move in order to accommodate the binding of G protein to activate its downstream signal pathway. In the **M1**-MOR complex (Figure 13a), TM3 and TM6 moved counter-clockwise at about 0.8 Å and 8.3 Å, respectively, and TM5 moved clockwise at about 3.2 Å. In **FW-AII-OH-2**-DOR (Figure 13b), TM3, TM5 and TM6 moved outward to about 4.2 Å, 5.5 Å and 7.5 Å, respectively. In **FW-AII-OH-1**-KOR (Figure 13c), TM3, TM5 and TM6 underwent movement of about 1.3 Å, 2.6 Å and 3.7 Å, respectively. All these movements in **M1**-MOR, **FW-AII-OH-2**-DOR and **FW-AII-OH-1**-KOR could make the cytoplasmic end generate enough space to accommodate the G protein.

### 2.11. Transmembrane Helix Movements of Opioid Protein upon Agonist Binding

The binding of the agonist to GPCRs caused the motion of the TMs and structural relaxation, which was the early stage in the activation of GPCRs. Corresponding with the agonizing effects of **M1**, **FW-AII-OH-2** and **FW-AII-OH-1**, the TM helices of MOR, DOR and KOR took different extents of motion. Taking the radius of gyration (Rg) as an example (Figure 14), the Rg of helices 1–7 in the **M1**-MOR complex gradually increased after 70.0 ns, and the Rg of apo-MOR decreased after about 80.0 ns (Figure 14a). In the **FW-AII-OH-2**-DOR system, the Rg of TM1–7 maintained growth after about 30.0 ns, while in the apo-DOR system, the Rg maintained a downward trend (Figure 14b). The Rg values of TM1–7 in the KOR-**FW-AII-OH-1** complex gradually increased after 10 ns and was maintained at around 21.4 Å until the end of the MD simulation, but as for the apo-KOR system, the Rg values decreased in the first 70 ns and was maintained at about 20.7 Å (Figure 14c). The results suggest that the binding of agonists to opioid receptors caused the structural relaxation of the receptors, contributing to the binding of G protein. The Rg values of the agonist-binding systems were higher than those of the ligand-free protein, indicating structural relaxation upon the binding of the agonist in the active state.

### 2.12. The Signal Transduction Mechanisms of MOR, DOR and KOR

The activation mechanisms of the three subtypes of opioid receptors were similar. Taking MOR as an example, upon the binding of selective MOR agonist **M1**, the protonated amino group in **M1** formed the salt bridge interaction with D^3.32^, and the benzene ring of M1 formed a π-π stacking with Y^7.43^. The 3–7 lock (hydrogen bond between D^3.32^ and Y^7.43^) was kept. The same phenomenon was also observed in our agonist-bound DOR system, and while the behavior of the 3–7 lock in the agonist-bound KOR was different from that of MOR and DOR, it was unlocked upon activation. Next, the DRG switch (interaction between R^3.50^ and T^6.34^) was fractured upon activation. It led to the rearrangement of I^3.40^, P^5.50^ and F^6.44^ with a counter-clockwise rotation (extracellular viewer). This rearrangement forced TM5 and TM6 to move clockwise at about 5.3 Å and 2.7 Å, respectively. TM3 slightly moved about 0.8 Å. All the processes above resulted in the structural relaxation of ORs to generate enough space to accommodate G protein in the intracellular site.

## 3. Materials and Methods

### 3.1. Chemistry

All reagents and solvents were purchased from Sinopharm Chemical Reagent Co., Ltd., Adamas-beta and other suppliers without further purification. The tetrahydrofuran was boiled with sodium and distilled before use. NMR data were recorded on a Bruker AMX-400 instrument. Chemical shifts (δ) are expressed in parts per million (ppm) relative to tetramethylsilane (TMS) as an internal standard. Mass spectra were measured on an Agilent 1100 series LC/MSD 1946D spectrometer with electric ionization (ESI). The purities of the tested compounds were determined by HPLC analysis and were ≥95%, which was considered to be pure enough for biological assays.

6-amino-3,4-dihydronaphthalen-1(2*H*)-one, **2**

Step 1: For preparation of 2-bromo-2-methylproanamide, a 250-mL, 3-necked, round-bottomed flask fitted with a thermometer, pressure-equalizing dropping funnel and equipped with a mechanical stirrer was charged with 28.0–30.0% aqueous sodium hydroxide (35.7 mL and approximately 260 mmol) and water (44.3 mL). The dropping funnel was charged with 2-bromo-2-methylpropanoyl bromide (50.0 g and 217.5 mmol), and the mixture was cooled to below 5 °C (internal temperature) using a brine/ice bath with stirring. The 2-bromo-2-methylpropanoyl bromide was added dropwise slowly with stirring so that the internal temperature did not rise above 15 °C with cooling using a brine/ice bath. Stirring was continued for 1 h after the addition was complete, with the temperature of the reaction mixture maintained between 0 and 5 °C. The resulting white precipitate was collected by filtration with a Brucella funnel and washed 3 times with 40 mL of water to give (33.0 g and 91.4%) a white crystalline powder, m.p. 147.3~148.5 °C.

Step 2: A 500-mL, 3-necked, round-bottomed flask fitted with a reflux condenser, thermometer and addition funnel and equipped with a large magnetic stirring bar was charged with 3,4-dihydro-6-hydroxy-1(2H)-naphthalenone (10.0 g and 61.7 mmol) and *N,N*-dimethylacetamide (90 mL). Sodium hydroxide (7.40 g and 185 mmol) was added into the flask by temporarily removing the addition funnel, and the resulting mixture was stirred at 20–30 °C for 1 h. Then, 2-bromo-2-methylpropanamide (30.7 g and 185 mmol) was added to the reaction mixture through the addition funnel, and the resulting mixture was stirred at 25–35 °C for 5 h. After the reaction period, sodium hydroxide (22.2 g and 555 mmol) was added, and the resulting mixture was stirred for 1 h with heating to 50–60 °C (internal temperature) using an oil bath. After the reaction was complete, water (90 mL) was added by using the addition funnel, and the mixture was heated at reflux using an oil bath for 1 h (internal temperature: 85–95 °C). Water (180 mL) was added to the reaction solution using the addition funnel, and the resulting precipitated mixture was allowed to cool slowly to 20–30 °C. The precipitated crystalline powder was collected by filtration with a Brucella funnel and washed 3 times with 90 mL of water to obtain a brown crystalline powder **2** (6.28 g and 60%), m.p. 128.8~131.2 °C.

6-bromo-3,4-dihydronaphthalen-1(2*H*)-one **3**

To the solution of **2** (5 g and 31 mmol) in a mixture of hydrobromic acid (50 mL and 47%) and water (50 mL) cooled by a brine/ice bath, a solution of sodium nitrite (2.5 g and 36 mmol) in water (20 mL) was added dropwise. The resulting mixture was stirred for 15 min before a solution of cuprous bromide (5.14 g and 36 mmol) in hydrobromic acid (20 mL and 47%) was added dropwise. The resulting mixture was then stirred for 15 min. The aqueous layers were washed with ethyl acetate (3 × 15 mL). The organic layers were combined, dried over Na_2_SO_4_, filtered and evaporated. The resulting yellow oil was purified by chromatography (petroleum ether:ethyl acetate = 40:1) to yield 6.03 g of **3** as a light yellow oil (83% yield).

6-phenyl-3,4-dihydronaphthalen-1(2*H*)-one **4**

To the solution of **3** (5.00 g and 20.82 mmol) in toluene, phenylboronic acid (2.79 g and 22.90 mmol) and potassium carbonate (5.76 g and 41.64 mmol), a catalytic amount of tetrakis(triphenylphosphine)palladium(0) (50 mg and 0.04 mmol) was added under nitrogen. The resulting mixture was then heated to reflux overnight. The reaction mixture was then filtered, evaporated under low pressure, and after chromatography (petroleum ether:ethyl acetate 20:1) to give 3.65 g of **4** as a white powder (75% yield), m.p. 105.7~107.3 °C.

6-(2-methoxyphenyl)-3,4-dihydronaphthalen-1(2*H*)-one **7**

Compound **7** was prepared according to the preparation of **4** as 4.65 g (83% yield) of white powder.

General Procedure A for the Preparation of **5** and **8**.6-phenyl-2-((methyl(phenethyl)amino)methyl)-3,4-dihydronaphthalen-1(2*H*)-one, **5**

A 100-mL round-bottomed flask fitted with a reflux condenser and thermometer and equipped with a magnetic stirring bar was charged with **4** (2.22 g and 10.0 mmol), *N*-methyl-2-phenylethan-1-amine (1.49 g and 11 mmol), paraformaldehyde (0.06 g and 2 mmol) and 25 mL isopropanol. The mixture was treated by hydrochloric acid to modify the pH to 3–4, heated at 90 °C for 15 min and had additional paraformaldehyde (0.06 g, 2 mmol) added 4 times before being reacted overnight. The resulting mixture was cooled to room temperature and evaporated. The residue was added to water (25 mL), extracted with ethyl acetate (3 × 10 mL), combined with the organic phase, washed with saturated salt water and then dried over Na_2_SO_4_, filtered and evaporated. The resulting light-yellow oil (2.59 g and 70% yield) was used in the next step without purification.

6-(2-methoxyphenyl)-2-((methyl(phenethyl)amino)methyl)-3,4-dihydronaphthalen-1(2*H*)-one, **8**

According to General Procedure A, yellow oil (1.60 g at 40.40%) was obtained and directly used in the next step without purification.

1-(3-hydroxyphenyl)-2-((methyl(phenethyl)amino)methyl)-6-phenyl-1,2,3,4-tetrahydronaphthalen-1-ol hydrochloride **FW-DIII-OH-2**

First, (3-bromophenyl)-tert-butyl-dimethylsilane (3.01 mL, 20.8 mmol) was dissolved in THF (17 mL) in an oven-dried round-bottomed flask under argon, and the resulting solution was cooled to −78 °C. BuLi (8.32 mL of a 2.5-M solution in hexanes at 20.8 mmol) was syringed in, and the resulting solution was stirred at −78 °C for 1 h. Then, **5** (1.53 g and 4.15 mmol) was syringed in, and the resulting solution was stirred at −78 °C for 2 h before adding 17 mL satd. aq. NH_4_Cl to the cold solution. This mixture was then warmed to room temperature, the solution was settled, and the aqueous phase was extracted using ethyl acetate (2 × 25 mL). The organic extracts were combined, dried over Na_2_SO_4_, filtered and concentrated under reduced pressure. The resulting yellow oil was dissolved in 34 mL DMF and 3 mL H_2_O, and then Cs_2_CO_3_ (676 mg and 2.08 mmol) was added into the solution. After stirring at room temperature for 1 h, the resulting mixture was diluted with 34 mL ethyl acetate and 17 mL H_2_O. The layers were separated, and the aqueous phase was extracted by ethyl acetate (2 × 25 mL). The organic extracts were combined, dried over by Na_2_SO_4_, filtered and concentrated under reduced pressure. The resulting residue was purified by chromatography (dichloromethane:methanol 20:1) to yield a yellow oil. The oil was dissolved in ether, and the hydrogen chloride-ether solution was added into the above solution to give a white precipitant, which was then filtered and dried to obtain 1.81 g of a white powder solid (yield 94.4%). ^1^HNMR (400 MHz, DMSO-d^6^) δ 9.20 (s, 1H), 7.68–7.59 (m, 2H), 7.50–7.31 (m, 5H), 7.26 (t, J = 7.3 Hz, 2H), 7.22–7.11 (m, 3H), 7.07 (t, J = 7.9 Hz, 1H), 6.93 (d, J = 8.2 Hz, 1H), 6.81 (s, 1H), 6.71 (d, J = 7.8 Hz, 1H), 6.59 (dd, J = 7.8, 2.0 Hz, 1H), 2.95–2.77 (m, 2H), 2.56 (t, J = 8.6 Hz, 2H), 2.49–2.32 (m, 3H), 2.24 (d, J = 11.3 Hz, 1H), 2.11 (d, J = 14.4 Hz, 4H), 2.00–1.86 (m, 1H), 1.76 (td, J = 15.9, 8.0 Hz, 1H). ^13^CNMR (400 MHz, CDCl3-d1) δ 156.20, 139.37, 136.76, 135.12, 130.46, 128.32, 126.12, 121.13, 118.86, 114.23, 110.53, 58.91, 58.66, 41.63, 35.30, 29.08, 28.33, 26.59, 24.91, 23.92, 22.07, 13.51. ESI-MS m/z 464.2 [M + H]^+^, HRMS (ESI) calculated for [C_32_H_33_NO_2_] 464.2584, finding 464.2588, m.p. 202.1~204.6 °C.

1-(3-hydroxyphenyl)-6-(2-methoxyphenyl)-2-((methyl(phenethyl)amino) methyl)-1,2,3,4-tetrahydronaphthalen-1-ol hydrochloride, **FW-DI-OH-2**

This is according to the preparation of **FW-DIII-OH-2**, and starting from **8** (0.20 g and 0.55 mmol) to produce **FW-DI-OH-2** (180mg, 72%). ^1^HNMR (400 MHz, DMSO-d^6^) δ 9.23 (s, 2H), 7.41–7.06 (m, 11H), 7.01 (t, *J* = 7.4 Hz, 1H), 6.91–6.71 (m, 3H), 6.61 (s, 1H), 5.80 (s, 1H), 3.75 (s, 3H), 2.85 (s, 3H), 2.14 (dd, *J* = 100.3, 67.6 Hz, 9H), 1.30–1.19 (m, 2H). ^13^CNMR (400 MHz, CDCl3-d1) δ 155.80, 155.19, 138.97, 136.95, 134.77, 130.26, 128.11, 126.08, 120.21, 118.86, 113.54, 110.53, 58.91, 58.66, 54.88, 41.63, 35.30, 29.08, 28.33, 26.59, 24.91, 23.92, 22.07. ESI-MS m/z 494.2 [M + H]^+^, HRMS (ESI) calculated for [C_33_H_35_NO_3_] 494.2690, finding 494.2698, m.p. 200.2~202.4 °C.

### 3.2. Homology Modeling of Human Active MOR

Because the human active DOR and KOR crystal structures were reported in 2019 [26] and 2018 [35], respectively, they were used in molecular simulations. While in terms of DOR, only the active state structure of Mus musculus MOR (PDB code: 5C1M) [30] was solved, this served as the template to construct the 3D structure of human active MOR. The amino acid sequences of human MOR were collected from the UniProtKB database (accession code: P35372). Sequence alignment of MOR was performed using the Discovery Studio Sequence Analysis program (BIOVIA, San Diego, CA, USA), and the sequence identity was 97.3% (the results for sequence alignment are listed in Appendix A). The homology modeling and loop refinement were optimized using Discovery Studio Modeler. Ten homology models were generated for MOR after loop refinement. Profiles-3D and Ramachandran plots were employed to evaluate the validity of the homology model of MOR (Appendix A).

### 3.3. Protein Preparation

The homology structure of human active MOR and the crystal structures of active DOR (PDB code: 6PT3), active KOR (PDB code: 6B73), inactive MOR (4DKL), inactive DOR (4N6H) and inactive KOR (4DJH) were imported into the Schrödinger software package. The protein structures were prepared with the Protein Preparation Tool in the Schrödinger package, and Asn, Gln and His residues were automatically checked for protonated states. Hydrogen atoms were added into the three structures at the physiology pH environment by the PROPKA tool in Maestro with an optimized hydrogen bond network.

### 3.4. Molecular Docking

The geometry of the probe ligands was built in Discovery Studio 3.5 and saved in the sdf file format. Optimization was performed at the DFT/B3LYP/6-311G** level. The constructed structures of the ligands were imported into Maestro and subjected to a Monte Carlo multiple minimum conformational search using the OPLS_2005 force field. Water was taken as an implicit solvent, and the output conformations of the ligands were used as the starting point for the docking experiments.

The docking pocket Grid files of the three ORs were generated at 20 Å around amino acid residue D^3.32^ with the Receptor Grid Generation module and docked with the Glide docking module (Glide 5.8) [36,37]. The Van der Waals scaling was set to 0.8 for the nonpolar atoms of the receptor and ligand. During docking, the number of docking outputs was set to 100 poses per docking run. The most reasonable conformation was selected for the next molecular dynamics simulations. The criteria to select the most reasonable conformation were the docking score, predicted binding energy, frequency at which docking conformation occurred, and all of the available experimental data.

### 3.5. Molecular Dynamics Simulations

The MD simulations were performed using the GROMACS package version 5.1.2 with the GROMOS96 force field [38,39] (www.gromacs.org (accessed on 15 February 2022)). The molecular topology files for the ligands were generated with PRODRG [40] (http://davapc1.bioch.dundee.ac.uk/prodrg/ (accessed on 15 February 2022)). The partial atomic charges of the ligands were determined using the CHelpG method [41] implemented in the Gaussian09 program [42] with the DFT/B3LYP/6-311G** basis set. For the MD simulations, all models were embedded in a hydrated POPC lipid bilayer. The procedure and parameters for constructing the receptor/hydrated POPC systems were similar to those used in our previous membrane protein simulations [43,44,45]. Figure 15 shows, taking the **M1**-MOR complex as an example, the structural model of the receptor/POPC/water systems. Before the MD simulations, the stepwise energy minimizations ((1) fix protein and optimize the rest of the parts, including lipids, ions and water; (2) fix water, ions and lipids, as well as the side chain of the opioid receptor and optimize the main chain of protein before optimizing the whole protein and (3) optimize the whole system without any restriction) were performed on the eight receptor/hydrated POPC systems first for all water molecules to remove their poor contacts with protein atoms and then for the whole system until the maximum force was no greater than 500 kJ^.^mol^−1^nm^−1^.

The solvent (water and POPC) molecules of each initial system were equilibrated with protein structures by constraining the solute (MOR, DOR and KOR) at 50, 100, 200 and 310 K for 500 ps. Then, the protein was equilibrated for 500 ps, and the solvent molecules were constrained at 50, 100, 200 and 310 K. Afterward, each system was equilibrated for 1 ns without any constraints. To maintain the systems at a constant temperature of 310 K, a Berendsen thermostat [46] was applied using a coupling time of 0.1 ps for the bulk water and POPC. The pressure was maintained by coupling to a reference pressure of 1 bar. The lengths of all bonds, including those of hydrogen atoms, were constrained by the LINCS algorithm [38]. Electrostatic interactions between the charged groups within 9 Å were calculated explicitly, and long-range electrostatic interactions were calculated using the particle mesh Ewald method [47] with a grid width of 1.2 Å and a fourth-order spline interpolation. A cutoff distance of 14 Å was applied for the Lennard-Jones interactions. Numerical integration of the equations of motion used a time step of 2 fs, with atomic coordinates saved every 1 ps for later analysis. To neutralize the modeled systems and simulate the physiological environment in humans, 46 Na^+^ and 59 Cl^−^, 46 Na^+^ and 61 Cl^−^, 46 Na^+^ and 51 Cl^−^, 46 Na^+^ and 60 Cl^−^, 46 Na^+^ and 60 Cl^−^, 46 Na^+^ and 52 Cl^−^, 46 Na^+^ and 60 Cl^−^ and 46 Na^+^ and 60 Cl^−^ ions were added to the molecular systems of the free MOR, DOR and KOR systems and the **M1**-MOR, **FW-AII-OH-2**-DOR, **FW-AII-OH-1-**KOR, **FW-DI-OH-2-DOR** and **FW-DIII-OH-2**-DOR complexes, respectively. Finally, 8 100-ns MD simulations were performed on these systems under the periodic boundary conditions in the NPT canonical ensemble. The apo-MOR (4DKL), apo-DOR (4N6H), apo-KOR (4DJH), **M1**-MOR, **FW-AII-OH-2**-DOR, **FW-AII-OH-1**-KOR, **FW-DI-OH-2**-DOR and **FW-DIII-OH-2**-DOR complexes were run through 100-ns MD simulations 3 times under the periodic boundary conditions in the NPT canonical ensemble.

### 3.6. Cluster Analysis

The 10,000 protein conformations extracted every 10 ps from the trajectory of the agonist-OR complex simulation systems were clustered based on the root mean square deviations (RMSDs) of the conformations using the GROMOS conformational cluster analysis method as implemented in GROMACS. A cutoff value of 1.0 Å was employed as the criterion to assign a cluster. The representative structure in the most popular cluster was used to analyze the selectivity to different opioid receptors.

### 3.7. Radio Ligand Binding Assay

Chinese hamster ovary (CHO) cells stably transfected with human KOR, DOR and MOR were obtained from SRI International (Palo Alto, CA, USA) and George Uh1 (NIDA Intramural Program, Bethesda, MD, USA). The cells were grown in 100-mM dishes in Dulbecco’s modified Eagle’s medium (DMEM) supplemented with 10% fetal bovine serum (FBS) and penicillin–streptomycin (10,000 U/mL) at 37 °C under a 5% CO_2_ atmosphere. The affinity and selectivity of the compounds for the multiple opioid receptors were determined by incubating the membranes with radiolabeled ligands and 12 different concentrations of the compounds at 25 °C in a final volume of 1 mL of 50 mM Tris-HCl with a pH of 7.5. Incubation times of 60 min were used for the MOR-selective peptide [^3^H]DAMGO, the KOR-selective ligand [^3^H]U69593 and the DOR-selective antagonist [^3^H]DPDPE.

### 3.8. [^35^S]GTP-γ-S Functional Assay

In a final volume of 0.5 mL, various concentrations of each tested compound were incubated with 7.5 mg (μ) of CHO cell membranes that stably expressed the human MOR. The assay buffer consisted of 50 mM Tris-HCl, pH 7.4, 3 mM MgCl_2_, 0.2 mM EGTA, 3 mM GDP and 100 mM NaCl. The final concentration of [^35^S]GTP-γ-S was 0.08 nM. Nonspecific binding was measured by the inclusion of 10 mM GTP-γ-S. Binding was initiated by the addition of the membranes. After an incubation period of 60 min at 30 °C, the reactions were terminated by rapid filtration, and the radioactivity was determined by liquid scintillation counting.

## 4. Conclusions

Based on the structure of **M1** (MOR, *K*_i_ = 13.0 ± 0.5 nM), an active metabolite of tramadol, we found two selective opioid ligands [22], which were **FW-AII-OH-2** (DOR, *K*_i_ = 4.7 ± 0.1 nM) and **FW-AII-OH-1** (KOR, *K*_i_ = 140 ± 9.0 nM). Through further structural modification, we obtained **FW-DIII-OH-2** (DOR, *K*_i_ = 228.5 ± 5.8 nM) and **FW-DI-OH-2** (DOR, *K*_i_ = 8.7 ± 0.9 nM). Taking these selective ligands as the molecular probes, we performed 100-ns molecular dynamic simulations to explore their selectivity against three OR subtypes. Finally, a stepwise activation mechanism of MOR, DOR and KOR induced by their corresponding agonist was proposed.

The main difference among the activation processes of MOR, DOR and KOR was the behavior of the 3–7 lock. In the MOR system, the 3–7 lock was broken from the inactive state to the active one because **M1** prevented the interaction between D^3.32^ and Y^7.43^, while in the DOR system, the 3–7 lock always existed from the inactive state to the active state because of the stable hydrogen bond network among D^3.32^, Y^7.43^ and the selective DOR ligand. In the KOR system, the 3–7 lock was formed in the active state and did not exist in the inactive state, because **FW-AII-OH-1** formed an interaction with D^3.32^ and pulled D^3.32^ toward Y^7.43^, promoting the formation of hydrogen bond interaction between D^3.32^ and Y^7.43^.

Y^7.43^ played a vital role in selectivity on the three opioid receptors. Upon the binding with KOR, the ligands formed a salt bridge interaction with D^3.32^ and a π-π stacking interaction with Y^7.43^ simultaneously, while upon binding with DOR, the ligands formed another hydrogen bond with Y^7.43^ or D^3.32^ to form a stable hydrogen bond network. Furthermore, W^6.58^ participated in adjusting the selectivity of DOR **FW-AII-OH-2**. **FW-DI-OH-2** and **FW-DIII-OH-2** formed a π-π stacking interaction with W^6.58^, while K^6.58^ (MOR) and E^6.58^ (KOR) were both hydrophilic amino acids, which could not form hydrophobic interactions.

## Data Availability

The presented data are available in this article and Appendix A.

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
