# Peer review of "Computational Methods for Understanding the Selectivity and Signal Transduction Mechanism of Aminomethyl Tetrahydronaphthalene to Opioid Receptors"

_molecules, 2022, doi:10.3390/molecules27072173_

Round 1

Reviewer 1 Report

  1. Introduction section should be modified with current studies about aminomethyl Tetrahy‐ 3 dronaphthalene related to this investigation. Specially, in silico computational study about aminomethyl Tetrahy‐ 3 dronaphthalene must be included.
  2. In section 2.1 “The Phenolic hydroxyl and N,N-dimethyl components have been considered as the essential group (“message” part) for tramadol and structure-related analogues to bind with opioid receptors  while the biphenyl group serves as the “address” part (Figure 3A)”. Why you have considered the phenolic hydroxyl and N,N-dimethyl components as the essential group? Clarify it.
  3. What about melting points of the synthesized compounds? The authors should determine it.
  4. The authors have performed molecular docking. But there is no data of non-bonding interactions. Add a summarize result of non-bonding interactions to clarify the key residues.
  5. Have you used any standard drug during in vitro test?
  6. In the section 3 (discussion) “The main difference among the activation processes of MOR, DOR and KOR was the behavior of 3-7 lock. In MOR system, the 3-7 lock was broken from inactive state to active one, while in DOR system, the 3-7 lock was always existed from inactive state to active state. In KOR system, the 3-7 lock was formed in the active state and didn’t exist in the inactive state”. Have you found any reason for this difference? Justify it.
  7. As this study related to drug-protein interactions, the authors should be added a Structure Activity Relationship (SAR) study for more clarification.
  8. Modify Figures 9, 11 and 14 as these are totally obscure.

Reviewer 2 Report

The work of Peng Xie et al., is a computational studies on M1 derivativesstructural determinants responsible for the selectivity of opioid receptor subtypes. It is a complex, detailed and well designed work.

I have only some concerns regarding the manuscript organization and results presentation. 

  1. I found several typing errors (e.g. double fulls top on line 45; the first discussion sentence is truncated, the verb is missing).
  2. there is no space between figure legends and text, thus, the reading is difficult at times.
  3. the figure 4 is missing. 
  4. In some figure captions there is a wrong or missing reference to the panel letter (e.g: in fig 5 the authors named panel a, b,b, d and e)

So I strongly recommend a deep revision of text, figure captions and layout of the work.

Moreover, the discussion paragraf is useless as it. I suggest to better discuss the results or to include the discussion in the result ("results and discussion" paragraph). Finally, I think that it will be better to put the conclusion after discussion and not after the methods section.

Round 2

Reviewer 1 Report

Accept